# Mental health literacy in nursing students: Insights from a cross-sectional analysis

Mohamad AlMekkawi[1], Rouwida ElKhalil[2], Annie Rosita Arul Raj[1], Ibrahim Bashayreh[1], Iffat Elbarazi[2], Mohammed Al Maqbali[1], Ciara Hughes[3]*

1 Nursing Department, Fatima College of Health Sciences, Abu Dhabi, United Arab Emirates, 2 Institute of Public Health, United Arab Emirates University, Al Ain, United Arab Emirates, 3 Institute of Nursing and Health Research, School of Health Sciences, Ulster University, Belfast, United Kingdom

* cm.hughes@ulster.ac.uk

## Abstract

Mental health literacy is crucial for nursing students to deliver effective patient care; however, its development throughout their academic journey remains underexplored. This study aimed to assess mental health literacy among nursing students in the UAE and examine factors influencing their literacy levels. A quantitative cross-sectional study was conducted from February 2024 to August 2024 using convenience sampling. A total of 295 undergraduate nursing students participated in the study. Data were collected using the Mental Health Literacy Questionnaire (MHLq) and analyzed using SPSS software (version 24). Descriptive and inferential analyses were conducted to calculate means, standard deviations, percentages, and measures of association using t-tests for students' sociodemographic variables, dimensions, and global scores, with a significance level of 0.05 for the tests. The findings indicated that participants (all female, with a mean age of $20.7 \pm 1.85$ years) had a mean MHLq score of $108.19 \pm 10.53$. Fourth-year students scored higher ($110.78 \pm 9.79$) than lower-year students ($106.85 \pm 10.68$). Students with family or friends who were affected had higher scores ($48.00 \pm 6.16$) than those without ($45.97 \pm 6.16$). The highest-scoring domain was knowledge of mental health problems ($46.59 \pm 6.40$), while self-help strategies scored lowest ($16.99 \pm 2.61$). Students with a personal history of mental illness had lower scores. The study also indicated a statistically significant association between students' marital status, their level of study, and their first-aid skills and help-seeking behavior. The study highlights the importance of integrating mental health literacy into undergraduate nursing curricula to enhance student's ability to provide patient-centered care for individuals with mental health disorders. Implementing targeted educational strategies focusing on awareness, recognition, and communication may strengthen students' competency and preparedness for mental health care practice.

**Data availability statement:** Data cannot be shared publicly because of the Rules Governing the Ethics of Scientific Research. Data are available from the Fatima College of Health Sciences Institutional Data Access / Ethics Committee (contact via fchs.ethics@fchs.ac.ae) for researchers who meet the criteria for access to confidential data.

**Funding:** The author(s) received no specific funding for this work.

**Competing interests:** The authors have declared that no competing interests exist.

## Introduction

The escalating prevalence of mental health illnesses across diverse communities, irrespective of their socioeconomic status, has emerged as a pressing global public health issue [1]. According to the World Health Organization (WHO), mental health is defined as the optimal state of physical and mental health in which individuals are aware of their capabilities, can effectively manage typical life pressures, can engage in productive and meaningful work, and can make valuable contributions to their communities [2]. At the same time, mental illness is a health condition characterized by changes in emotion, thinking, behavior, or a combination of these [2]. Globally, youths are experiencing mental illnesses, which account for a significant portion of the global disease burden in this age group. The WHO estimates that one in seven adolescents (ages 10–19) has a mental illness, making issues like depression, anxiety, and behavioral disorders leading causes of illness and disability among adolescents. Suicide ranks as the fourth leading cause of death among 15–29-year-olds, highlighting the critical need for addressing mental health conditions early in life to prevent long-term consequences [3].

Nursing students are particularly susceptible to a broad spectrum of mental health illnesses, often experiencing heightened stress and anxiety relative to their peers in other disciplines [4]. Their academic obligations, relationship challenges, and financial concerns are all contributing factors to students' mental health illnesses [5]. The COVID-19 pandemic has also exerted a notable impact on health sciences students' mental health, including post-traumatic stress disorder, insomnia, depression, anxiety, and stress symptoms [6]. A systematic evaluation conducted in China among healthcare students, including medical, nursing, and pharmacy students, revealed a significant frequency of common mental illnesses. The prevalence of depression was 29%, anxiety was 18%, suicidal thought was 13%, suicide attempt was 3%, and suicide plan was 4% [7]. In addition, these students face severe stigma when seeking help for mental health illnesses [8]. Due to stigma, they prefer to seek support from friends and family rather than professional help [9].

Numerous studies were carried out to understand the factors contributing to the high prevalence of mental illnesses [1,10]. One of the identified factors contributing to the high prevalence of mental illnesses was the perceived low level of Mental Health Literacy (MHL). The potential of MHL to empower individuals with knowledge and beliefs about mental illnesses that aid in their recognition, management, or prevention offers a beacon of hope in this scenario [10]. MHL evolved from health literacy and was described by Jorm et al. as knowledge and beliefs about mental illnesses that aid in their recognition, management, or prevention [11]. Mental health research has shown that improving MHL enhances the early detection of mental illnesses, improves outcomes and the use of health services, and facilitates community action to promote better mental health [12]. Promoting MHL among undergraduate nursing students can improve attitudes toward seeking help and increase help-seeking intentions, minimizing stigma and facilitating the utilization of help-seeking services [13].

Mental health in the Arab world faces unique challenges, including limited research and mental health literacy, even among healthcare professionals. However, lower

suicide rates are reported compared to global averages. Factors such as strong family bonds and religious beliefs provide some protective effects [14]. On the other hand, stigma and cultural perceptions around mental illness often prevent individuals from seeking help. Addressing mental health needs in the Arab world requires a multifaceted approach, including improving mental health literacy, addressing stigma, and enhancing access to care [14]. The majority of the population in the United Arab Emirates (UAE) consists of young people. Studies indicate that young adults experiencing psychological and emotional distress, potential indicators of emerging mental health illnesses, are less inclined to pursue professional assistance or access mental health services than their older counterparts [15]. Nursing students in the region often receive insufficient mental health education, with curricula lacking comprehensive training and offering limited opportunities for clinical exposure to mental health settings [5,16]. This highlights the significance of increasing awareness of mental illness, enhancing culturally sensitive intervention initiatives, and developing strategies to improve access to mental health services in the region [17,18].

Additionally, recent research shows that healthcare professionals in the UAE who work with individuals with mental health illnesses have low MHL and need urgent attention [16]. Therefore, this study aimed to investigate mental health literacy among nursing students in the UAE. The objectives were to investigate the level of MHL among nursing students in the UAE and to explore the influence of sociodemographic factors on the dimensions of MHL.

## Materials and methods

### Study design, population, and setting

A cross-sectional study was conducted to collect data from a convenience sample of undergraduate nursing students at the Fatima College of Health Sciences (FCHS) in the UAE. Convenience sampling was selected due to its practicality and feasibility, given the study's time and resource constraints. All students (N = 1200) across four campuses (Abu Dhabi, Al Ain, Ajman, and Al Dhafra) were invited to complete an online survey from February 28, 2024, to August 30, 2024. Two hundred ninety-five students completed the survey, with a response rate of 24.5% of the total population.

FCHS was founded in 2006 to become the leading provider of health education in the United Arab Emirates. Initially, the college provided a Bachelor of Science (BSc) in Nursing program for Emirati female students. Due to the high demand for healthcare professionals in the UAE, FCHS launched four bachelor's degree programs in 2011. These programs include pharmacy, physiotherapy, radiography, medical imaging, and emergency health services. In 2019, the college introduced a psychology program, followed by a midwifery program in 2022. The college has 2,000 students enrolled in six undergraduate health sciences programs located at four campuses in different geographical areas of the UAE: Abu Dhabi, Al Ain, Al Dhafra, and Ajman.

### Procedures

The study used the Mental Health Literacy Questionnaire (MHLq) developed by Dias et al. in 2018 [19]. The questionnaire was sent to participants via an official e-mail from the college registry. The research team obtained permission from the college to access students' lists and e-mail addresses. The email contained a clear subject line and information explaining the study's aim and the survey duration, prompting recipients to click on the hyperlink to launch their web browser and access the questionnaire. The questionnaire took approximately 30 minutes to complete. All participants were prompted to consent to the questionnaire once the questionnaire was started.

### Measures

The MHLq was initially developed and validated for use among young adults in Western contexts [19]. The MHLq has demonstrated strong reliability and validity across various settings, including among healthcare students [20,21]. However, its cultural appropriateness for the UAE context requires careful consideration. To ensure the questionnaire's relevance

and sensitivity to the local context, the research team conducted a preliminary review of the MHLq items, focusing on cultural and linguistic appropriateness. No significant cultural barriers were identified, as the items primarily address universal concepts related to mental health knowledge, beliefs, and behaviors. The questionnaire was administered in its original English form, as all participants at the college were required to demonstrate a high level of English proficiency before enrolling in the nursing program. Moreover, the research team reviewed the questionnaire to ensure that the language and examples used were culturally neutral and aligned with local values and religious beliefs. A pilot test was conducted with a small group of nursing students (n = 20) to evaluate the clarity and cultural appropriateness of the questionnaire. Feedback from the pilot test indicated that the items were well understood and relevant to the student's experiences. No further modifications were considered necessary.

The MHLq is a valid and reliable tool for identifying gaps in knowledge, beliefs, and behavioral intentions regarding MHL [19]. The authors reported acceptable to good reliability scores for the dimensions (Knowledge of mental health problems = 0.74; Erroneous beliefs/stereotypes = 0.72; First aid skills and help-seeking behavior = 0.71; Self-help strategies = 0.60), as well as a global score of 0.84. For this study, the questionnaire demonstrated good internal consistency with Cronbach's alpha of 0.783 for all 29 items. The questionnaire has two sections. The first section consists of participants' demographic information, which includes the age, gender, marital status, specialty, level of study, campus, proximity to a person with mental illness, history of mental illness, and source of mental health information. The second section comprises 29 items, each using a 5-point Likert scale (1 = Strongly disagree, 2 = Disagree, 3 = Neither agree nor disagree, 4 = Agree, and 5 = Strongly agree). The items are divided into four dimensions: Knowledge of mental health problems (11 items), Erroneous beliefs or stereotypes (8 items), First aid skills and help-seeking behavior (6 items), and Self-help strategies (4 items). The total score ranges between 29 and 145, with higher values in the dimensions, and the global score corresponds to higher levels of MHL [19].

### Data analysis

The study data were downloaded from the electronic survey into IBM SPSS Statistics software version 29. Descriptive and inferential analyses were conducted to calculate means, standard deviations, percentages, and measures of association using t-tests for students' sociodemographic variables, dimensions, and global scores, with a significance level of 0.05 for the tests.

### Ethics statement

The Study Obtained Ethics Approval from the FCHS Ethics Committee (Reference No: FCEC-2-22-23-BSN-1-SF). Data collection started after obtaining ethical approval. Participants were provided with an explanation of the study's purpose, assurances of voluntary participation, protection of their privacy, and the option to withdraw. Informed consent was obtained from all participants before they participated in the survey. Upon accessing the survey link, participants encountered an informed consent statement that detailed the study's objectives, voluntary participation, confidentiality protocols, and their right to withdraw at any time. Participants were permitted to advance to the survey only after actively consenting to the statement, thereby ensuring informed participation. Given that the survey was conducted online, consent was obtained electronically and documented within the survey platform.

### Results

Table 1 presents a summary of the sociodemographic variables of the study. Two hundred ninety-five nursing students participated, with no missing values in the scale items. All participants were female (100%), and the majority (34.2%) were in their fourth year of study across four different regions of the UAE: Abu Dhabi (18.6%), Ajman (39%), Al Ain (30.5%), and Al Dhafra (11.9%). Participants were between 20 and 25 years old, with a mean age of 20.7 years (SD = 1.85).

**Table 1. Sociodemographic variables of students.**

| Variables (N = 295) | n | % |
|---|---|---|
| **Gender** | | |
| Female | 295 | 100 |
| **Marital Status** | | |
| Married | 15 | 5.1 |
| Single | 280 | 94.9 |
| **Level of Study** | | |
| Year 1 | 73 | 24.7 |
| Year 2 | 67 | 22.7 |
| Year 3 | 54 | 18.3 |
| Year 4 | 101 | 34.2 |
| **Campus** | | |
| Alain | 90 | 30.5 |
| Abu Dhabi | 55 | 18.6 |
| Ajman | 115 | 39.0 |
| Al Dhafra | 35 | 11.9 |
| **Have been diagnosed with a mental health disorder** | | |
| No | 253 | 85.8 |
| Yes | 42 | 14.2 |
| **Have any family member or friend with a mental health disorder** | | |
| No | 204 | 69.2 |
| Yes | 91 | 30.8 |
| **Source of information about mental health** | | |
| Books and Journals | 162 | 54.9 |
| Family member | 75 | 25.4 |
| Friends | 90 | 30.5 |
| The Nurse | 90 | 30.5 |
| The Doctor | 142 | 48.1 |
| Mental healthcare professional | 178 | 60.3 |
| Internet | 200 | 67.8 |
| Social media | 188 | 63.7 |
| **Age** | **Mean** | **SD** |
| | 20.47 | 1.85 |

*SD: standard deviation*

Regarding students' experiences with mental health, only 14.2% reported having been previously diagnosed with a mental health disorder. When asked about their proximity to individuals with mental health disorders, 30.8% indicated that they have a friend or family member affected by such conditions. Additionally, participants were questioned about their sources of information on mental health; 67.8% cited the internet as their primary source, 63.7% mentioned social media, and 60.3% referred to mental health professionals.

Table 2 presents the mean scores and standard deviations for the MHLq's four dimensions and global scores. Higher results across all domains and for the global score indicate higher levels of MHL. The MHLq global score was 108.19 (SD = 10.53), indicating a reasonably high level of MHL among the students. The knowledge of mental health problems domain scored 46.59 (SD = 6.40), erroneous beliefs or stereotypes scored 20.30 (SD = 4.33), first aid skills and help-seeking behavior 24.29 (SD = 4.05), and self-help strategies 16.99 (SD = 2.61).

**Table 2. MHLq scores among students.**

| MHLq Dimensions | Mean | Scale Score Range | SD |
|---|---|---|---|
| Knowledge of mental health problems (11 items) | 46.59 | 11-55 | 6.40 |
| Erroneous beliefs or stereotypes (8 items) | 20.30 | 8-40 | 4.33 |
| First aid skills and help-seeking behavior (6 items) | 24.29 | 6-30 | 4.05 |
| Self-help strategies (4 items) | 16.99 | 4-20 | 2.61 |
| Global Score (29 items) | 108.19 | 29-145 | 10.53 |

*SD: standard deviation*

Table 3 shows the items which had the highest scores: "Mental disorders affect people's thoughts" (M = 4.54, SD = 0.72), "If someone close to me had a mental disorder, I would listen to her/him without judging or criticizing" (M = 4.52, SD = 0.72), "If someone close to me had a mental disorder, I would encourage her/him to look for a psychologist" (M = 4.45, SD = 0.75), and "One of the symptoms of depression is the loss of interest or pleasure in most things" (M = 4.43, SD = 0.82). Students scored lower on the items "If I had a mental disorder, I would seek friends' help" (M = 3.53, SD = 1.12) and "If I had a mental disorder, I would seek my relatives' help" (M = 3.64, SD = 1.13).

Table 4 summarizes the association between the participants' demographic variables, MHLq dimensions, and global scores. Students' knowledge of mental health problems (Dimension 1) was significantly affected by their level of study and the proximity of people with mental health illnesses. Students enrolled in the fourth year of their study (M = 49, SD = 5.56) and those in proximity of people with mental health illnesses (M = 48, SD = 6.16) had higher mean scores than students at lower levels of study (M = 45.37, SD = 6.48), and students who had no proximity to people with mental health illnesses (M = 45.97, SD = 6.16).

The study also revealed a statistically significant association between students' level of study and erroneous beliefs or stereotypes (Dimension 2), with first-, second-, and third-year students (M = 20.98, SD = 4.45) scoring higher than fourth-year students (M = 19.01, SD = 3.80). Additionally, a statistically significant association was found between students' marital status and level of study, as well as their first-aid skills and help-seeking behavior (Dimension 3). Fourth-year students in both dimensions (3 and 4) (M = 25.27, SD = 3.57) and (M = 17.55, SD = 2.37) scored higher than the first-year, second-year, and third-year students (M = 23.79, SD = 4.21) and (M = 16.71, SD = 2.69), and married students (M = 26.27, SD = 2.66) scored higher than single students (M = 24.19, SD = 4.09) in first aid skills and help-seeking behavior. Finally, the fourth-year students (M = 110.78, SD = 9.79) had higher global scores than those in their first, second, and third years of study (M = 106.85, SD = 10.68), indicating that students' level of study has a significant impact on their global scores.

**Table 3. Students' highest and lowest item scores.**

| Scale Items | Mean | SD |
|---|---|---|
| *Highest Scores* | | |
| 25. Mental disorders affect people's thoughts. | 4.54 | 0.72 |
| 11. If someone close to me had a mental disorder, I would listen to her/him without judging or criticizing. | 4.52 | 0.72 |
| 5. If someone close to me had a mental disorder, I would encourage her/him to look for a psychologist. | 4.45 | 0.75 |
| 20. One of the symptoms of depression is the loss of interest or pleasure in most things. | 4.43 | 0.82 |
| *Lowest Scores* | | |
| 18. If I had a mental disorder, I would seek friends' help. | 3.53 | 1.12 |
| 4. If I had a mental disorder, I would seek my relatives' help. | 3.64 | 1.13 |

*SD: standard deviation*

**Table 4. Association between demographic variables and MHLq dimensions.**

| MHLq Dimensions | | Marital Status | | Student Level of Study | | Family/Friend with Mental Illness | | History of Mental Health Illness | |
|---|---|---|---|---|---|---|---|---|---|
| | | Married | Single | <Yr. 4 | Yr. 4 | Yes | No | Yes | No |
| Knowledge of mental health problems | N | 15 | 280 | 194 | 101 | 91 | 204 | 42 | 253 |
| | M (SD) | 46.00 (4.75) | 46.63 (6.49) | 45.37 (6.48) | 48.95 (5.56) | 47.99 (6.75) | 45.97 (6.16) | 45.50 (6.03) | 46.77 (6.46) |
| | P | .713 | | <.001** | | .012* | | .233 | |
| Erroneous beliefs or stereotypes | N | 15 | 280 | 194 | 101 | 91 | 204 | 42 | 253 |
| | M (SD) | 19.87 (4.17) | 20.33 (4.35) | 20.98 (4.45) | 19.01 (3.80) | 20.08 (4.63) | 20.41 (4.20) | 20.76 (5.65) | 20.23 (4.09) |
| | P | .686 | | <.001** | | .541 | | .465 | |
| First aid skills and help-seeking behavior | N | 15 | 280 | 194 | 101 | 91 | 204 | 42 | 253 |
| | M (SD) | 26.27 (2.66) | 24.19 (4.09) | 23.79 (4.21) | 25.27 (3.57) | 24.29 (4.09) | 24.30 (4.05) | 23.60 (4.10) | 24.41 (4.05) |
| | P | .053* | | .003* | | .979 | | .228 | |
| Self-help strategies | N | 15 | 280 | 194 | 101 | 91 | 204 | 42 | 253 |
| | M (SD) | 17.20 (1.70) | 16.99 (2.66) | 16.71 (2.69) | 17.55 (2.37) | 17.07 (2.65) | 16.97 (2.60) | 16.64 (2.25) | 17.06 (2.67) |
| | P | .758 | | .008* | | .762 | | .345 | |
| Global Score | N | 15 | 280 | 194 | 101 | 91 | 204 | 42 | 253 |
| | M (SD) | 109.33 (5.15) | 108.13 (10.75) | 106.85 (10.68) | 110.78 (9.79) | 109.42 (10.96) | 107.65 (10.32) | 106.50 (9.30) | 108.47 (10.72) |
| | P | .668 | | .002* | | .183 | | .262 | |

**. P-value is significant at the 0.01 level (2-tailed) – Independent sample T-test.

*. P-value is significant at the 0.05 level (2-tailed) – Independent sample T-test.

SD: standard deviation. M: mean.

## Discussion

The study findings highlight several key insights into the MHL among nursing students. The students' global score was 108.19±10.53, which was higher than the scores reported by Dias et al., who had a score of 103.93±7.10 [19]. However, this score was lower than the findings from other studies, such as Kristina et al.'s research in Indonesia, which revealed a global score of 113.25±12.22 [22], and Campos et al.'s study involving samples from China, India, Indonesia, Portugal, Thailand, and the United States [23].

The study investigated the influence of sociodemographic factors on the MHLq dimensions and overall scores. Students' academic levels were positively associated with higher MHL scores. Fourth-year students achieved higher scores than those in their second and third years. These findings align with other research that identified a significant association between academic year and MHL scores among students [22]. Similarly, Saito and Creedy, and Jafari et al. found significant differences between academic levels, with second and third-year students reporting significantly higher mean MHL scores [12,24]. The higher scores may be due to the fact that senior nursing students had either completed or were nearing completion of a relevant mental health nursing course. Furthermore, these students gained hands-on experience by providing care to patients with mental health conditions during their clinical rotations at mental health facilities. Additionally, students with experience in mental illness achieved higher scores compared to their peers without such experience. These findings are in line with Dias, Campos (19), which indicated that participants who reported knowing someone with mental health issues scored higher in all dimensions of the MHLq than those lacking acquaintances facing such

challenges. In a similar vein, Chao, Lien (21) noted that higher levels of MHL were linked to decreased social distance from individuals with mental illness. This study also found that students with a history of mental disorders did not show increased levels of MHL, which contradicts previous findings by Saito and Creedy and Gorczynski et al. [12,25].

The study also highlighted that the reported sources of information about mental health varied among students, with the internet and social media as primary resources closely followed by mental health professionals. This distribution suggests a reliance on digital platforms for information, which the quality and accuracy of the accessed content may influence the quality and accuracy of the content accessed [26–28]. The COVID-19 pandemic has significantly affected mental health globally, resulting in a 25% increase in the prevalence of anxiety and depression [29]. The pandemic spawned an "info-demic" of credible and non-credible health information, creating hurdles for individuals seeking accurate guidance. This scenario highlights the necessity of mental health literacy in navigating misinformation, recognizing mental health issues, and seeking appropriate care during such unprecedented times [30].

Notably, the COVID-19 pandemic has deeply impacted students' mental health overall, intensifying stress, anxiety, and depression due to the sudden transition to remote learning, social isolation, and the hurdles posed by continuing their studies under pandemic conditions, all of which have affected their mental well-being and potentially their mental health literacy. Nursing students were better positioned to access health-related information [31,32]. These factors shaped their mental health literacy, underscoring the importance of accessible and reliable mental health information and support services during times of crisis. Mental health literacy encompasses the ability to recognize and comprehend mental disorders, as well as the capability to prevent these disorders, seek appropriate education and treatment, employ self-help techniques, and offer support to individuals experiencing mental health challenges [33]. The lack of knowledge and the stigma associated with mental illnesses present significant barriers for individuals seeking help and accessing mental health services [34]. Therefore, nursing students and nurses should learn to assess patients' mental health literacy needs, utilize the healthcare system effectively, and communicate vital health information clearly to enhance patient and community health outcomes.

## Limitations

While this study offers valuable insights into mental health literacy among nursing students, it is essential to acknowledge certain limitations. First, using cross-sectional and self-reporting tools might introduce response bias and inherently limit the generalizability of the findings. Second, the data was collected non-randomly from four campuses of the same institution, and only female students were included. To improve the study's generalizability and cultural adaptations, it would have been beneficial to include broader or more diverse populations from different geographical locations, institutions, and genders in the data collection. Finally, a high non-response rate may introduce non-response bias, as participants who did not respond might have been more confident in their MHL, potentially leading to an overestimation of MHL levels in the sample. Therefore, future research should consider a larger-scale study or a longitudinal or mixed-methods approach, as integrating qualitative insights can provide a deeper understanding of student's experiences and perceptions, further informing the development of effective educational interventions.

## Conclusions

The study concludes that nursing students have a relatively high level of mental health literacy, particularly in their knowledge of mental health problems, with strengths in recognizing symptoms and promoting professional help. However, they are less likely to seek help from friends or family if they experience a mental disorder. Mental health literacy is significantly influenced by years of study and personal proximity to individuals with mental health challenges. Nursing students, those in advanced years, and students with close connections to individuals with mental health issues demonstrated more substantial knowledge and skills, particularly in first aid, help-seeking, and self-help strategies. As such, this study emphasizes the need to enhance MHL among nursing students through targeted

curriculum development, clinical training, and initiatives aimed at reducing stigma. Nursing programs should incorporate comprehensive mental health education that emphasizes help-seeking behaviors and self-care strategies while expanding clinical rotations in mental health settings to provide more hands-on experience. Each clinical course in the nursing curriculum should integrate and emphasize its mental health components. For instance, pediatric, maternity, geriatric, and palliative care each have distinct mental health aspects that require dedicated attention rather than being condensed into just a few slides. Awareness campaigns and peer support programs can help mitigate stigma and normalize discussions surrounding mental health. Institutional policies should prioritize student well-being by ensuring access to counseling services and mental health resources. Faculty training on MHL can further support these initiatives. These actions will not only enhance students' mental health literacy but also empower them to deliver high-quality, patient-centered care to individuals with mental health disorders, ultimately contributing to improved mental health outcomes in the UAE and beyond.

## Supporting information

**S1 Text. Mental Health Literacy Questionnaire (MHLq).** 10.6084/m9.figshare.28662296.
(CSV)

## Acknowledgments

The authors express their gratitude to all study participants for their cooperation and support, as well as to those who contributed to the completion of this manuscript in various capacities.

## Author contributions

**Conceptualization:** Mohamad AlMekkawi, Rouwida ElKhalil, Annie Rosita Arul Raj, Ibrahim Bashayreh, Iffat Elbarazi, Mohammed Al Maqbali, Ciara Hughes.

**Data curation:** Mohamad AlMekkawi, Rouwida ElKhalil, Annie Rosita Arul Raj, Ibrahim Bashayreh, Iffat Elbarazi, Mohammed Al Maqbali, Ciara Hughes.

**Formal analysis:** Rouwida ElKhalil, Annie Rosita Arul Raj, Iffat Elbarazi, Mohammed Al Maqbali.

**Funding acquisition:** Mohamad AlMekkawi, Rouwida ElKhalil.

**Investigation:** Rouwida ElKhalil.

**Methodology:** Mohamad AlMekkawi, Rouwida ElKhalil, Annie Rosita Arul Raj, Ibrahim Bashayreh, Iffat Elbarazi, Mohammed Al Maqbali, Ciara Hughes.

**Project administration:** Rouwida ElKhalil.

**Resources:** Rouwida ElKhalil.

**Software:** Rouwida ElKhalil, Iffat Elbarazi.

**Supervision:** Rouwida ElKhalil.

**Validation:** Rouwida ElKhalil.

**Visualization:** Rouwida ElKhalil.

**Writing – original draft:** Mohamad AlMekkawi, Rouwida ElKhalil, Annie Rosita Arul Raj, Ibrahim Bashayreh, Iffat Elbarazi, Mohammed Al Maqbali, Ciara Hughes.

**Writing – review & editing:** Mohamad AlMekkawi, Rouwida ElKhalil, Annie Rosita Arul Raj, Ibrahim Bashayreh, Iffat Elbarazi, Mohammed Al Maqbali, Ciara Hughes.

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
