## [Decision Letter · Decision Letter 0]

24 Mar 2025

PONE-D-25-06755Mental Health Literacy in Nursing Students: Insights from a Cross-Sectional AnalysisPLOS ONE

Dear Dr. Hughes,

Thank you for submitting your manuscript to PLOS ONE. After careful consideration, we feel that it has merit but does not fully meet PLOS ONE’s publication criteria as it currently stands. Therefore, we invite you to submit a revised version of the manuscript that addresses the points raised during the review process.

We look forward to receiving your revised manuscript.

Kind regards,

Fatma Refaat Ahmed, Ph.D.

Academic Editor

PLOS ONE

Reviewers' comments:

Reviewer's Responses to Questions

**Comments to the Author**

1. Is the manuscript technically sound, and do the data support the conclusions?

Reviewer #1: Yes

Reviewer #2: Yes

Reviewer #3: Yes

2. Has the statistical analysis been performed appropriately and rigorously? 

Reviewer #1: Yes

Reviewer #2: Yes

Reviewer #3: Yes

3. Have the authors made all data underlying the findings in their manuscript fully available?

Reviewer #1: Yes

Reviewer #2: Yes

Reviewer #3: Yes

4. Is the manuscript presented in an intelligible fashion and written in standard English?

Reviewer #1: Yes

Reviewer #2: Yes

Reviewer #3: Yes

5. Review Comments to the Author

Reviewer #1: Dear authors

Your topic and your manuscript are very good, but in the abstract in method, please explain the analysis method, and in the result, please add the association finding.

In conclusion, add some sentences for practical using of your finding.

Best regards.

Reviewer #2: This is an interesting paper and the content is very good but I suggest the paper can be improved in the following ways: Abstract

-Please correct all parts of the article according to the guidelines of the journal authors guideline In the methods section please bring year of performing of this study, sampling methods and data analysis methods -In the conclusion part, it is necessary to specify the researcher's proposal to improve the conditions and use of the beneficiaries

Introduction

Please bring the following items 1- Definition of the research problem 2- The magnitude and importance of the study variable 3- Expressing the necessity of conducting the study Finally, the practical purpose of the study should be stated. Methods

Please report the details scoring and validations of study tools

Discussion

In the discussion section, it is necessary y to compare the main results of the study with the results of other studies in this field. To strengthen the article, especially in the introduction and discussion section the following studies are suggested, please used and add to this manuscript references.

-Medical tourism development: A systematic review of economic aspects

- The effect of bio ethical principles education on ethical attitude of prehospital paramedic personnel

- Investigating the burden of disease dimensions (time-dependent, developmental, physical, social and emotional) among family caregivers with COVID-19 patients

Conclusion � What are your suggestion for future studies? Best regard

Reviewer #3: This topic is very crucial. It is important for nursing students to promote mental health awareness and address mental health issues as future healthcare providers. The researchers had limitation in the sample size " 295" students only.

6. PLOS authors have the option to publish the peer review history of their article (what does this mean? ). If published, this will include your full peer review and any attached files.

**Do you want your identity to be public for this peer review?** For information about this choice, including consent withdrawal, please see our Privacy Policy .

Reviewer #1: No

Reviewer #2: No

Reviewer #3: **Yes: ** Ola Mousa

---

## [Author Response · Author response to Decision Letter 1]

28 Mar 2025

We have addressed each of the reviewers’ points in detail and made the necessary revisions to the manuscript. A point-by-point response is provided in the attached file titled "Responses to Reviewer."

---

## [Decision Letter · Decision Letter 1]

15 Apr 2025

Mental Health Literacy in Nursing Students: Insights from a Cross-Sectional Analysis

PONE-D-25-06755R1

Dear Dr. Hughes,

We’re pleased to inform you that your manuscript has been judged scientifically suitable for publication and will be formally accepted for publication once it meets all outstanding technical requirements.

Kind regards,

Fatma Refaat Ahmed, Ph.D.

Academic Editor

PLOS ONE

Additional Editor Comments (optional):

Reviewers' comments:

Reviewer's Responses to Questions

**Comments to the Author**

1. If the authors have adequately addressed your comments raised in a previous round of review and you feel that this manuscript is now acceptable for publication, you may indicate that here to bypass the “Comments to the Author” section, enter your conflict of interest statement in the “Confidential to Editor” section, and submit your "Accept" recommendation.

Reviewer #2: (No Response)

2. Is the manuscript technically sound, and do the data support the conclusions?

Reviewer #2: (No Response)

3. Has the statistical analysis been performed appropriately and rigorously? 

Reviewer #2: (No Response)

4. Have the authors made all data underlying the findings in their manuscript fully available?

Reviewer #2: (No Response)

5. Is the manuscript presented in an intelligible fashion and written in standard English?

Reviewer #2: (No Response)

6. Review Comments to the Author

Reviewer #2: Dear authors Many thanks for your good response

Best regards

Dear authors Many thanks for your good response

Best regards

7. PLOS authors have the option to publish the peer review history of their article (what does this mean? ). If published, this will include your full peer review and any attached files.

**Do you want your identity to be public for this peer review?** For information about this choice, including consent withdrawal, please see our Privacy Policy .

Reviewer #2: No

---

## [Editor Report · Acceptance letter]

PONE-D-25-06755R1

PLOS ONE

Dear Dr. Hughes,

I'm pleased to inform you that your manuscript has been deemed suitable for publication in PLOS ONE. Congratulations! Your manuscript is now being handed over to our production team.

Kind regards,

on behalf of

Dr. Fatma Refaat Ahmed

Academic Editor

PLOS ONE